# Changes in the Work Schedule of Nurses Related to the COVID-19 Pandemic and Its Relationship with Sleep and Turnover Intention

**DOI:** 10.3390/ijerph19148682

**Published:** 2022-07-17

**Authors:** Ingebjørg Louise Rockwell Djupedal, Ståle Pallesen, Anette Harris, Siri Waage, Bjørn Bjorvatn, Øystein Vedaa

**Affiliations:** 1Department of Psychosocial Science, University of Bergen, 5015 Bergen, Norway; staale.pallesen@uib.no (S.P.); anette.harris@uib.no (A.H.); siri.waage@uib.no (S.W.); oystein.vedaa@fhi.no (Ø.V.); 2Department of Health Promotion, Norwegian Institute of Public Health, 5018 Bergen, Norway; 3Norwegian Competence Center for Sleep Disorders, Haukeland University Hospital, 5021 Bergen, Norway; bjorn.bjorvatn@uib.no; 4Department of Global Public Health and Primary Care, University of Bergen, 5015 Bergen, Norway

**Keywords:** shift work, quick return, sleep, turnover intention, nurse

## Abstract

Background: This study aimed to investigate whether different types of changes in the work schedule of nurses working rotating shifts during the COVID-19 pandemic were associated with sleep duration, sleep quality, and turnover intention. Methods: Cross-sectional questionnaire data from 694 nurses participating in the SUrvey of Shift work, Sleep and Health (SUSSH) were collected between the first and the second wave of the COVID-19 pandemic in Norway. A total of 89.9% were female, and mean age was 44.6 years (SD = 8.6 years). Changes in the shift work schedule related to the pandemic comprised reports of more long workdays (>8 h), less days off between work periods, more night shifts, more quick returns (i.e., 11 h or less between two consecutive shifts), more day shifts, and more evening shifts compared to no change in the respective shift characteristics. Change in sleep duration, sleep quality, and turnover intention as well as demographics were also assessed. Logistic regression analyses were performed to investigate whether changes in the specific work schedules were associated with sleep duration, sleep quality, and turnover intention, controlling for sex, age, cohabitation, children living in household, percentage of full time equivalent and other changes in the work schedule. Results: A total of 17% reported experiencing one or more changes in their work schedule during the pandemic. Experiencing any change in the work schedule predicted worse sleep quality (OR = 2.68, *p* < 0.001), reduced sleep duration (OR = 4.56, *p* < 0.001), and higher turnover intention (OR = 1.96, *p* = 0.006) compared to experiencing no change in work schedule. Among the specific changes in work schedules, experiencing an increase in quick returns had the highest odds ratio for worse sleep quality (OR = 10.34, *p* = 0.007) and higher turnover intention (OR = 8.49, *p* = 0.014) compared to those who reported no change in quick returns. Nurses experiencing an increase in long workdays were more likely to report higher turnover intention (OR = 4.37, *p* = 0.003) compared to those experiencing no change in long workdays. Conclusions: Change in work schedule related to the pandemic was associated with worse sleep quality, reduced sleep duration, and higher turnover intention. Increase in quick returns emerged as especially problematic in terms of sleep quality and turnover intention, along with long workdays, which were associated with higher turnover intention.

## 1. Introduction

The worldwide outbreak of the COVID-19 pandemic has affected billions of people’s lives. The extent of its implications is not yet fully known, but the pandemic has undoubtedly put a significant strain on healthcare systems across the world. Healthcare workers, including nurses, play a critical role in the preparedness and management of the pandemic, which in turn may have led to an increasingly demanding work situation and heavy workload [1]. It is reasonable to expect that healthcare workers have experienced changes in their work schedule at short notice during the pandemic due to urgent need for increased staffing and to cover absence of other employees (e.g., work absence due to closed schools/kindergartens, or quarantine or isolation after close contact with the virus). This likely represents a burden that comes in addition to the challenges healthcare workers already face, such as the strains of being engaged in shift work and its associated consequences regarding health and safety [2].

Shift work is a way of organizing working hours in order to maintain a service beyond the normal daytime and is characterized by employees succeeding each other at the workplace [3]. Working outside the normal daytime working hours can cause a disturbance of the individuals’ endogenous circadian rhythm, which is believed to be an important contributor to the detrimental effects of shift work [2,4]. Studies on shift work have found associations with health problems such as certain cancers and cardiovascular diseases [5,6], diabetes [7], gastro-intestinal problems [8], and impaired mental health [9], as well as more acute problems related to sleep and stress [2,10]. Shift systems that cause the greatest disruption of the biological rhythm and sleep—in particular night shifts, early morning shifts (starts before 06:00 a.m.) and quick returns (i.e., 11 h or less between two consecutive shifts)—are associated with the largest negative effects on health and sleep [11,12,13,14]. These shifts are thus typically considered most unfavorable. The most common and immediate consequence of shift work is disturbances of sleep. Studies have uncovered a high prevalence of insufficient sleep (70.1%), insomnia (36.0%), and self-reported short sleep duration (≤6 h) (48.6%) among nurses working shifts [15], as well as frequent reports of poor sleep quality (78%) [16]. Recent studies conducted during the pandemic show a similar high prevalence of sleep disturbances [17] and insomnia symptoms [18,19,20] among healthcare workers. Furthermore, a recent study from Norway utilizing the same data as in the present study found that the majority of nurses reported no change in sleep duration (84.2%), sleep quality (82.4%), or sleep patterns from before to during the pandemic (data collected between the first and the second pandemic wave) compared to retrospective accounts of their sleep before the pandemic [21]. This might suggest that the pandemic had limited impact on the already high prevalence of sleep disturbances among nurses, at least at the time of data collection. Nonetheless, it should be noted that a small but still significant proportion reported shorter sleep duration (11.9%) and worse sleep quality (15.9%) during the pandemic [21]. This raises concern and warrants further investigation.

It is conceivable that the COVID-19 pandemic may have led to an increase in the proportion of healthcare workers considering leaving their job [22]. Turnover intention reflects the intention to quit one’s current job, and although it does not necessarily lead to actual turnover, it is a strong predictor thereof [23]. Nurses have reported significantly higher turnover intention during the COVID-19 pandemic compared to before the pandemic [24]. Large proportions of frontline nurses caring for COVID-19 patients have reported high intention to leave their current health care setting (29.5%) and profession (22.3%) [25]. More than 17% of newly educated nurses in the United States are reported to leave their first job within the first year [26], and turnover entails major costs and can also impair the quality of nurse care and patient outcomes [27,28]. There is ample research on factors that predict turnover intention [29]. Although shift work is assumed to be a prominent reason for turnover intention, there is still limited evidence for a specific role of shift work in turnover and turnover intention [29]. Furthermore, no previous research has looked at whether changes in work schedule, for example due to a pandemic, have affected nurses’ turnover intention.

Against this backdrop, we aimed to investigate whether different types of changes in the work schedule of nurses working rotating shifts during the COVID-19 pandemic were associated with sleep duration, sleep quality, and turnover intention.

## 2. Materials and Methods

### 2.1. Procedure and Participants

We used data from the SUrvey of Shift work, Sleep and Health (SUSSH), which has been conducted among Norwegian nurses since 2008/2009. In this survey, a random sample of 5400 nurses from the Norwegian Nurses Organization’s (NNO) membership roll was invited to participate (wave 1). This first subsample consisted of five equal strata based on the number of years since completing their degree in nursing (0–11 months, 1.1–3 years, 3.1–6 years, 6.1–9 years, 9.1–12 years). In total, 2059 nurses (response rate = 38.1%) from this subsample completed the questionnaire. In an effort to increase the study population, a new subsample of 906 newly graduated nurses (response rate = 33.1%) was recruited in 2009, resulting in a total baseline cohort of 2964 nurses. This cohort has been invited to answer annual follow-up questionnaires, except for nurses who have withdrawn from the study, died, or moved to an unknown address.

The present study utilizes data collected as a pandemic wave of the SUSSH cohort in 2020. A questionnaire focusing on sleep and pandemic-related variables was developed and sent by postal mail (together with a prepaid return envelope). In addition, the nurses were given the opportunity to complete the questionnaire online using a unique identifier. The questionnaire was sent out the first week of June 2020. Up to two reminders were sent to those who did not respond before the data collection was closed after 18 weeks. A total of 1532 nurses responded to the questionnaire (1454 paper version, 78 internet version), resulting in a response rate of 55%. The aim of this study was to focus on nurses working rotating shifts. That is, only those who (1) were currently working as nurses, (2) worked rotating two-shifts or three-shifts (e.g., day and evening shifts or day, evening, and night shifts, respectively), and (3) had at least a 50% of full-time equivalent position were included in the analyses, totaling a sample of 694 nurses. A further rationale for including only those who worked rotating shifts was to isolate this particular group, so that, for example, the analyses regarding an increase in quick returns actually comprised a reference group that could have quick returns in their schedule.

The data collection took place from 1 June to 1 October 2020, between the first and the second wave of the COVID-19 pandemic in Norway. The first wave reached its peak around the end of March, with 60 hospitalizations due to COVID-19 on 25 March, compared to 0 admissions on 1 July (mid data collection). The second wave occurred late autumn 2020, peaking at 25 COVID-related hospital admissions on 4 December [30]. The total burden of measures imposed by the government followed the pandemic waves, and there were few intervening measures during the data collection period.

### 2.2. Instruments

#### 2.2.1. Demographic Characteristics

*Marital status* (married, partner, or cohabitating: yes/no) and *caretaker responsibility for children in the household* (children living at home: yes/no) were included as sociodemographic variables. Work-related variables such as whether they were *currently working as a nurse*, *working in a rotating shift schedule* with either two-shifts or three-shifts, and *percentage of full time equivalent* were also included in the questionnaire.

#### 2.2.2. Change in Work Schedule

*Change in work schedule* was assessed with one question asking if the nurses had changed their working schedule due to the COVID-19 pandemic (yes/no). *Change in work shifts* was identified with the question “If your work schedule has changed after the pandemic hit Norway, how would you describe these changes?”, with the following responses: “More long workdays (more than 8 h)”, “Less days off between work periods”, “More night shifts”, “More quick returns (less than 11 h of rest between two consecutive shifts)”, “More day shifts”, and “More evening shifts”. Respondents could report one or more of the listed work shift changes, and thus each listed change was coded either as “change in the respective shift” or “no change in the respective shift”.

#### 2.2.3. Sleep

*Change in sleep duration* was computed from two questions about sleep duration before and after the pandemic hit Norway: “Before the COVID-19 pandemic came to Norway, approximately how long did you sleep per day?” and “After the COVID-19 pandemic came to Norway, approximately how long do you sleep per day?” The response alternatives to both questions were “Less than 5 h”, “5–5.9 h”, “6–6.9 h”, “7–7.9 h”, “8–8.9 h”, “9–9.9 h”, “10 h or more”. Responses were then dichotomized into “no change or increased” and “reduced” sleep duration. *Change in sleep quality* was derived from the following question: “After the pandemic hit Norway, I sleep…” (1) “Much poorer than before”, (2) “To some degree poorer than before”, (3) “No change”, (4) “To some degree better than before”, and (5) “Much better than before”, with responses dichotomized into “equal or better” and “worse” sleep quality.

#### 2.2.4. Turnover Intention

*Turnover intention* was measured with the question “Has the COVID-19 pandemic led you to consider quitting as a nurse when the crisis is over?” (yes/no/don’t know). Responses were dichotomized into “yes” and “no/don’t know”.

### 2.3. Ethical Consideration

The study was approved by the Regional Committee for Medical and Health Research Ethics of Western Norway (REK-West, no. 088.08), and conducted in accordance with relevant guidelines and regulations. All participants included in the study provided informed written consent.

### 2.4. Statistical Analysis

Analysis was conducted using IBM SPSS 27 for Windows. Descriptive statistics are presented with means (M) and standard deviations (SD) for continuous variables and frequencies (*n*) and percentages (%) for categorical variables. To investigate the relationship between changes in work schedule and change in sleep duration, change in sleep quality, and turnover intention, separate logistic regression analyses were performed. Nurses were divided into different groups based on type of change in work schedule (i.e., no change in the respective shift characteristic vs. change in the respective shift characteristic). In previous research, night shifts and quick returns often merge as the two shift characteristics associated with most negative outcomes. A separate group that included those who reported an increase in night shifts and/or quick returns was therefore defined. Crude models were run (Model 1), as well as models adjusting for sex, age, cohabitation, children living in the household, and percentage of full time equivalent (Model 2). We extended this adjustment in model 3, additionally including the other changes in the work schedule (i.e., increase in the number of long workdays [>8 h], fewer days off between work periods, increase in night shifts, increase in quick returns, increase in evening shifts, and increase in day shifts). The aforementioned confounders were included as they define some basic elements of an individual’s life and work situation, all of which can conceivably account for some of the variation in person’s sleep and turnover intention. Results are presented as odds ratios, with 95% confidence intervals (CI). An alpha level of 0.05 was set to indicate statistical significance. Missing values were treated as invalid in the analyses.

## 3. Results

Descriptive statistics of demographics, sleep factors, and turnover intention among nurses in this study are reported in Table 1. Overall, there were no major differences in demographic characteristics between the nurses who experienced no change and those who experienced any change in their work schedule. Of the total sample, 89.9% of nurses were female and the mean age was 44.6 years (standard deviation (SD) = 8.6 years). In terms of cohabitation, 78.3% reported living with a partner, and most of the nurses (68.2%) had no children living at home. Two-thirds of the nurses (66.0%) reported working at least 90% of full time equivalent.

In terms of sleep variables, most of the nurses reported having equal or better sleep quality (83.6%) after the pandemic hit Norway compared to pre-pandemic time (82.3% reported having equal and 1.3% reported having better sleep quality). Change in sleep duration showed the same pattern in responses, with 88.4% of the nurses reporting having equal or increased sleep duration compared to before the pandemic (84.4% reported no change and 4.0% reported increase in sleep duration). Of the nurses included in this study, 9.7% reported considering quitting as a nurse when the crisis is over.

Table 2 presents an overview of the proportion of nurses who reported different changes in their work schedule during the COVID-19 pandemic. A total of 17.3% of the nurses reported experiencing change in their work schedule due to the pandemic. The most frequent reported change in work shifts was increased number of long workdays (>8 h) (*n* = 54), followed by increase in evening shifts (*n* = 53), fewer days off between work periods (*n* = 43), increase in quick returns (*n* = 43), increase in night shifts (*n* = 39), and increase in day shifts (*n* = 23).

Results from the logistic regression analysis are reported in Table 3. Collectively, the results show that experiencing a change in the work schedule during the pandemic was associated with increased odds of reporting worse sleep quality (OR = 2.68, *p* < 0.001, model 2, fully adjusted model), reduced sleep duration (OR = 4.56, *p* < 0.001, model 2, fully adjusted model), and turnover intention (OR = 1.96, *p* = 0.006, model 2, fully adjusted model) compared to experiencing no change in the work schedule. When inspecting the specific changes in the work shifts more closely, it was found that those who experienced an increase in long workdays (>8 h) during the pandemic were four times more likely to also report turnover intention (OR = 4.37, *p* = 0.003, model 3, fully adjusted model) compared to experiencing no change. Furthermore, the nurses who experienced an increase in quick returns were 10 times more likely to report worse sleep quality (OR = 10.34, *p* = 0.007, model 3, fully adjusted model), and 8 times more likely to report turnover intention (OR = 8.49, *p* = 0.014, model 3, fully adjusted model) compared to experiencing no change. Reporting an increase in night shifts and/or quick returns was associated with a higher risk of worse sleep quality, also when adjusting for other shift changes (OR = 4.65, *p* = 0.019, model 3, fully adjusted model) compared to experiencing no change in these shift characteristics. However, experiencing an increase in night shifts only was not associated with worse sleep quality, nor any of the other outcome variables. In addition, experiencing fewer days off between work periods, increase in day shifts, or increase in evening shifts were not associated with worse sleep quality, reduced sleep duration, or turnover intention in the fully adjusted model.

## 4. Discussion

The aim of this study was to examine how change in nurses’ work schedules during the COVID-19 pandemic was associated with sleep duration, sleep quality, and turnover intention. Overall, the results showed that experiencing any change in the work schedule was associated with worse sleep quality, reduced sleep duration, and turnover intention compared to no change in the work schedule. Examination of the changes in the specific shift schedule characteristics showed that it was mainly an increase in quick returns that explained worsening of sleep quality and the increase in turnover intention. However, experiencing an increase in long workdays (>8 h) was also significantly associated with turnover intention.

In the fully adjusted model, we found that those who experienced an increase in quick returns after the pandemic hit Norway had more than 10 times higher odds of reporting worse sleep quality, compared to those who reported no change in quick returns. This is in line with previous research showing detrimental effects of quick returns [31,32]. A cohort study found that the frequency of quick returns was a strong determinant for poor sleep quality, short sleep, dissatisfaction with work hours, and increased exhaustion [31]. Furthermore, a diary study found that quick returns from evening to day shifts were associated with worse sleep quality compared to sleep quality between two evening shifts, and that sleep duration associated with a quick return was two hours shorter than between two evening shifts [32]. The present study adds to these findings in that it also shows that changes in shift schedule in the form of more quick returns during the pandemic were associated with worse sleep quality. However, it was somewhat surprising that having more quick returns was not associated with shorter sleep duration. There may be several reasons for these results. For example, we do not precisely know what an “increase” in the shift characteristics entails, both in terms of magnitude and the pre-change value (e.g., the baseline number of quick returns). Still, regardless of how large the change in number of quick returns was, this study demonstrates that a change was associated with worse sleep quality, but not necessarily self-reported shorter sleep duration. Other factors that may help explain the lack of significant findings regarding sleep duration include the fact that there were few participants who experienced changes in shift schedule with the pandemic (small effects are more difficult to detect). In addition, participants were asked to provide a retrospective account of their sleep duration before the pandemic, which might be difficult due to a recall bias. In addition, it cannot be ruled out that actual sleep loss associated with quick returns could be compensated with more sleep at other points in the work schedule/free days, resulting in an overall status quo.

In an attempt to increase the number of observations in the analyses, we investigated whether the combined increase in quick returns and night shifts was associated with negative outcomes. We found that experiencing an increase in quick returns and/or night shifts was associated with more than four times higher odds of reporting worse sleep quality, albeit experiencing an increase in night shifts alone did not show significant results. This indicates that an increase in quick returns seems to be the most prominent adverse change in shift schedule among the two. These results support previous findings that have shown that the frequency of quick returns may be a stronger negative predictor compared to night shifts in terms of sleep and satisfaction with work hours [31].

In addition to worse sleep quality, results showed that those who experienced an increase in quick returns also had more than eight times higher odds of reporting turnover intention compared to those who had no change in quick returns. A similar result was also obtained for long workdays (>8 h), in which those who experienced an increase in long workdays had more than four times higher odds of reporting turnover intention. A total of 18% of those who experienced any change in their shift schedule during the pandemic reported that they considered quitting as a nurse after the pandemic, which is more than twice that of those who did not experience any change in their shift schedule. Previous studies have shown that turnover intention has increased among nurses during the pandemic [25], and the present investigation points to certain pandemic-related changes in the shift schedule that may be relevant in this context. Quick returns and long shifts (10 h or longer) in the work schedule have previously been found to represent a larger problem for shift workers than night shifts [33]. In line with the present study, an early pandemic study of healthcare workers in China found that long working hours (>8 h) were associated with turnover intention [34]. In this study, the reporting of an increase in long workdays and quick returns may represent the occurrence of unplanned overtime due to heavy workloads and shortages in staffing. These shift characteristics can conceivably represent an additional burden to an already heavy and demanding workload and serve as an explanation as to why these specific shift characteristics appear to be particularly unfavorable in terms of turnover intention. Furthermore, it is conceivable that the potentially increased risk of infection to which nurses are exposed in the workplace may have had an impact on sleep and turnover intention. In this study, we do not have specific information on the extent to which the nurses were exposed to an increased risk of being infected at the workplace. However, results from the study conducted by [21], which utilized the same data as in the current study, showed that only 1.2% (*n* = 15) of the nurses in the sample reported to have been infected by the coronavirus. The data collection was conducted at a time of low infection rates in Norway [30], and may explain the corresponding low infection rates reported by the study sample. During this period, there was a reopening of society where many of the measures imposed by the government during the first pandemic wave were reversed. These factors may elucidate why most of the nurses in the sample reported no change in sleep duration, sleep quality, and no turnover intention related to the pandemic.

To the best of our knowledge, this is the first study to investigate possible consequences of changes in work schedules of nurses related to the pandemic. A pandemic might require a greater need for staffing at some hospital units or an urgent need to cover absence of other employees due to quarantine, isolation, or closed schools/kindergartens, etc. Experiencing changes in shift schedule may further exacerbate the already known adverse health problems associated with shift work. Thus, it is important to understand how changes in work schedule might affect healthcare workers’ health and retention in the profession, which ultimately might also affect the quality of care. Such knowledge is useful since it provides insight into how best to deal with an urgent need for increased preparedness and staffing in the healthcare sector, the possible caveats of various work schedule changes, and where it is necessary to intervene to mitigate any possible negative effects.

### Strengths and Limitations

This study has some limitations that should be highlighted. The study population includes a homogeneous group of mainly female nurses, which limits the generalizability of findings across genders and other occupational groups. However, the skewness in terms of gender is representative of the study population, and the homogeneity of the sample can thus be said to be a strength of the study. Furthermore, only nurses who reported working rotating shifts (two-shifts or three-shifts) were included in the analysis, which means that we should be careful when generalizing the results to those who are engaged in day work only or other types of shift work. However, we do not have information on whether they worked forward- or backward-rotating shifts. Nonetheless, a strength of the study is the fairly large sample size and high response rates in the follow-up data. The cross-sectional design of the study limits the possibility to make inferences about causality; nevertheless, the questions were formulated in such a way that it allowed us to make temporal assumptions about the relationship between changes in work schedule, sleep, and turnover intention. Additional limitations include the implications of recall bias, as the study only included self-reported measures of sleep quality and sleep duration before and during the pandemic, as well as the common method bias [35]. Additionally, an important limitation is that sleep and turnover intention were measured using single item questions that have not previously been validated, which makes the validity of the measures somewhat uncertain. COVID-19 infection rates were low at the time of the survey, which may have had a bearing on how the nurses responded. Regarding reports on changes in the shift schedule, it is unclear how large the reported change was and what the pre-change values were (i.e., baseline and change in frequency of the specific shift characteristics in question). Furthermore, it is unclear whether similar changes also take place under more normal circumstances and how large an impact those would have. In this paper, we have considered some of the work schedule changes that may occur in order to solve challenges in the staffing of a hospital in the short term. Overall, only 17.3% (*n* = 120) of the respondents reported any change in the work schedule, with the most frequent change being increase in long workdays (>8 h) (*n* = 54). Despite the limitation of relatively small subgroups, significant results revealed that any change in the shift schedule, especially an increase in quick returns or long shifts, was a strong predictor for worse sleep quality and turnover intention.

## 5. Conclusions

In summary, the current study demonstrated that experiencing any change in the work schedule during the pandemic was associated with worse sleep quality, reduced sleep duration, and turnover intention. Experiencing an increase in quick returns proved to be the most adverse change in the shift schedule, displaying strong associations with worse sleep quality and turnover intention. The results also revealed that experiencing an increase in long workdays (>8 h) predicted turnover intention. Knowledge about adverse changes in work schedules is important to enable employers and employees to plan health-promoting work schedules that preserve the health of healthcare workers and their retention in the profession. Future studies should aim to use objective register data to investigate actual changes in nurses’ work schedules during the pandemic that can separate between hospital wards (i.e., frontline nurses vs. others), and investigate, among others, possible consequences for sleep, turnover intention, sickness absence, and susceptibility to infection.

## Figures and Tables

**Table 1 ijerph-19-08682-t001:** Demographics and characteristics among nurses (*n* = 694).

	No Change in Work Schedule		Any Change in Work Schedule		Total Sample	
	*n* (%)	Mean (SD)	*n* (%)	Mean (SD)	*n* (%)	Mean (SD)
**Descriptive information**						
Sex						
Female	523 (91.4)		99 (82.5 *)		624 (89.9)	
Male	48 (8.4)		19 (15.8)		67 (9.7)	
Age		44.6 (8.6)		45.3 (7.9)		44.8 (8.5)
Cohabitation						
Living with partner	445 (78.3)		94 (78.3)		541 (78.4)	
Living without partner	123 (21.7)		26 (21.7)		149 (21.6)	
Children living in household						
No children	384 (67.3)		87 (73.1)		472 (68.2)	
Child/children in household	187 (32.7)		32 (26.9)		220 (31.8)	
Percentage of full time equivalent						
50–75%	67 (11.7)		18 (15.0)		86 (12.4)	
76–90%	137 (24.0)		13 (10.8)		150 (21.0)	
>90%	368 (64.3)		89 (74.2)		458 (66.0)	
**Outcome variables**						
Sleep quality dichotomized						
Equal or better	494 (86.4)		85 (70.8)		580 (83.6)	
Worse	78 (13.6)		35 (29.2)		114 (16.4)	
Sleep duration dichotomized						
No change or increased	524 (91.9)		87 (72.5)		612 (88.4)	
Reduced	46 (8.0)		33 (27.5)		80 (11.6)	
Turnover intention						
Yes	45 (7.9)		22 (18.3)		67 (9.7)	
No	472 (83.1)		88 (73.3)		562 (81.4)	
Don’t know	51 (9.0)		10 (8.3)		61 (8.8)	

Note. SD, standard deviation. The table presents descriptive information from those who worked rotating shifts (two-part shifts (e.g., day and evening shifts) or three-part shifts (e.g., day, evening, and night shifts), had at least a 50% position, and who reported that they worked as a nurse at the time the survey was conducted. Change in the work schedule in this context could refer to whether the nurses have had an increase in long workdays (more than 8 h), fewer days off between work periods, increase in night shifts, increase in quick returns, increase in evening shifts, or increase in day shifts. * Two females reported missing for any change in the work schedule.

**Table 2 ijerph-19-08682-t002:** Overview of how many participants reported experiencing changes in their work schedule during the COVID-19 pandemic (*n* = 694).

	*n* (%)
**Have you experienced a change in your work schedule due to the COVID-19 pandemic?**	
Yes	120 (17.3)
No	572 (82.4)
Missing data	2 (0.3)
**If your work schedule has changed after the pandemic hit Norway, how would you describe these changes? ***	
Increase in long workdays (more than 8 h)	54 (7.7)
Fewer days off between work periods	43 (6.1)
Increase in night shifts	39 (5.6)
Increase in quick returns	43 (6.1)
Increase in day shifts	23 (3.3)
Increase in evening shifts	53 (7.6)
Increase in night shifts and/or quick returns	68 (9.7)

Note. The table shows descriptive information from those who worked rotating shifts two-shifts (e.g., day and evening shifts) or three-shifts (e.g., day, evening, and night shifts), had at least a 50% position, and who reported that they worked as a nurse at the time the survey was conducted. * Participants could report change in more than one shift work characteristic.

**Table 3 ijerph-19-08682-t003:** Results from logistic regression analyses on the association between change in work schedule and change in sleep quality, change in sleep duration, and turnover intention (*n* = 694).

	Crude Model		Adjusting for Background Variables ^a^		Adjusting for Background Variables ^a^ and Other Changes in Work Schedule ^b^	
	OR (95% CI)	*p*-Value	OR (95% CI)	*p*-Value	OR (95% CI)	*p*-Value
**Any change in the work schedule due to the pandemic ^c^**						
Worse sleep quality	2.70 (1.70–4.29)	**<0.001**	2.68 (1.67–4.32)	**<0.001**		
Reduced sleep duration	4.51 (2.72–7.48)	**<0.001**	4.56 (2.71–7.65)	**<0.001**		
Turnover intention	1.84 (1.16–2.93)	**0.010**	1.96 (1.22–3.16)	**0.006**		
**Increase in long workdays (>8 h)**						
Worse sleep quality	3.21 (1.54–6.66)	**0.002**	3.30 (1.56–7.01)	**0.002**	3.35 (1.21–9.33)	0.200
Reduced sleep duration	3.74 (1.66–8.43)	**0.001**	3.56 (1.56–8.13)	**0.003**	1.26 (0.28–5.72)	0.762
Turnover intention	3.74 (1.85–7.53)	**<0.001**	3.89 (1.91–7.91)	**<0.001**	4.37 (1.66–11.50)	**0.003**
**Fewer days off between work periods**						
Worse sleep quality	2.74 (1.26–5.98)	**0.011**	2.75 (1.24–6.12)	**0.013**	0.77 (0.09–6.37)	0.805
Reduced sleep duration	5.03 (2.25–11.24)	**<0.001**	4.59 (2.03–10.42)	**<0.001**	1.21 (0.15–10.13)	0.858
Turnover intention	4.41 (2.13–9.15)	**<0.001**	4.75 (2.25–10.01)	**<0.001**	3.21 (0.74–13.97)	0.121
**Increase in night shifts**						
Worse sleep quality	5.15 (2.38–11.14)	**<0.001**	6.35 (2.83–14.26)	**<0.001**	1.66 (0.19–14.83)	0.650
Reduced sleep duration	3.61 (1.46–8.92)	**0.005**	3.94 (1.53–10.13)	**0.005**	0.00 (0.00–0.00)	0.999
Turnover intention	2.58 (1.16–5.72)	**0.020**	2.86 (1.26–6.52)	**0.012**	1.10 (0.12–9.76)	0.931
**Increase in quick returns**						
Worse sleep quality	8.27 (4.03–16.98)	**<0.001**	11.35 (5.08–25.36)	**<0.001**	10.34 (1.91–56.09)	**0.007**
Reduced sleep duration	8.13 (3.84–17.20)	**<0.001**	8.09 (3.64–17.96)	**<0.001**	1.69 (0.18–15.80)	0.645
Turnover intention	4.19 (2.04–8.60)	**<0.001**	5.19 (2.42–11.12)	**<0.001**	8.49 (1.55–46.58)	**0.014**
**Increase in day shifts**						
Worse sleep quality	2.16 (0.68–6.86)	0.193	2.37 (0.70–7.97)	0.165	6.81 (0.86–54.27)	0.070
Reduced sleep duration	3.89 (1.21–12.58)	**0.023**	3.31 (0.99–11.09)	0.052	2.96 (0.29–30.47)	0.361
Turnover intention	0.69 (0.16–3.10)	0.632	0.73 (0.16–3.36)	0.689	0.00 (0.00–0.00)	0.999
**Increase in evening shifts**						
Worse sleep quality	4.00 (1.92–8.32)	**<0.001**	4.47 (2.08–9.58)	**<0.001**	5.96 (0.93–38.43)	0.060
Reduced sleep duration	2.95 (1.22–7.15)	**0.017**	3.06 (1.24–7.56)	**0.016**	0.00 (0.00–0.00)	0.999
Turnover intention	3.48 (1.70–7.13)	**<0.001**	3.92 (1.87–8.20)	**<0.001**	4.15 (0.65–26.75)	0.134
**Increase in quick returns and/or night shifts**						
Worse sleep quality	4.85 (2.59–9.09)	**<0.001**	6.20 (3.14–12.24)	**<0.001**	4.65 (1.29–16.75)	**0.019**
Reduced sleep duration	3.94 (1.91–8.12)	**<0.001**	3.92 (1.83–8.42)	**<0.001**	0.87 (0.10–7.23)	0.895
Turnover intention	2.94 (1.55–5.58)	**<0.001**	3.44 (1.76–6.73)	**<0.001**	3.35 (0.94–11.93)	0.062

Note. OR, odds ratio; CI, confidence interval. ^a^ Background variables include sex, age, cohabitation, children living in the household, percentage of full time equivalent. ^b^ Other changes in the work schedule include whether the nurses have had an increase in long workdays (more than 8 h), fewer days off between work periods, increase in night shifts, increase in quick returns, increase in evening shifts, or increase in day shifts. In the analysis of each of the specific work schedule changes, the model is adjusting for all the other respective shift changes. ^c^ The fully adjusted model for this regression is presented in Model 2.

## Data Availability

The datasets used and/or analyzed during the current study are available from the corresponding author upon reasonable request.

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
