# Peer review of "Changes in the Work Schedule of Nurses Related to the COVID-19 Pandemic and Its Relationship with Sleep and Turnover Intention"

_ijerph, 2022, doi:10.3390/ijerph19148682_

Round 1

Reviewer 1 Report

Thank you for the opportunity to review your article. The study is very interesting.

I suggest some integrations explained in the following specific comments:

1.      No text alignment; please correct

2.      The manuscript was not submitted in accordance with the requirements of the journal. It is difficult to comment when it is not possible to indicate lines of text where corrections are needed;

3.      I suggest that the sample of subjects be presented in the table;

4.      The third page of the manuscript contains the number "7" in the middle of the text in section 2.1. please correct;

5.      Page number 4 - twice written Turnover intention; please correct

6.      Research instrument - why did you not use standardized questionnaires to assess sleep quality and turnover ?

7.      Part of results:

Table 1.

Age not provided.Please corect.

Sample size was 694 and in the Table 1 indicating sex, cohabitant,children and other characteristics there are errors in the column- (n(%) ). The description of Table 1 does not correspond to the data provided. For example, 68.2% in the text and 67.3% in the table 1.

After Table 1 and Table 2 are notes– SD? but only n and % are given in the tables. Please correct. 

Author Response

Dear Editor,

Thank you for the invitation to submit a revised version of our paper to the Special Issue on Evidence-Based Effects of Irregular Working Hours on Physical and Mental in the International Journal of Environmental Research and Public Health.

We greatly appreciate the comments from the editor and reviewers. We have followed-up on these comments to the best of our ability and believe that the paper has improved significantly after these adjustments. With this revision, we provide point-by-point answers to all comments (please see attachment). Please note that comments from the editor / reviewers are highlighted in bold, while the author's responses are highlighted in italics and with indentation.

In addition, we include a 'clean' version of the manuscript, as well as a version that uses 'track-changes' to highlight any new text that has been added in response to the comments by editor / reviewers.

We would also like to make the editor aware that on our previous submission, most part of Table 1 was not uploaded properly during the submission process and there were also some issues with the formatting. Hence, it seems that the reviewers have considered a somewhat incomplete version of our submission. Several of the comments from the reviewers relate to the fact that the manuscript they received was incomplete. We therefore kindly ask the editor to ensure that our full submission is sent to the reviewers in this revision round.

Thank you for your time and consideration!

Kind regards,

Ingebjørg Louise Rockwell Djupedal

(on behalf of all authors)

Reviewer 2 Report

ijerph-1778017

The work is devoted to a topical issue, the study of the impact of shift work on the sleep function of nurses in the context of the COVID-19 pandemic. The advantage of this work is that it was started before the pandemic and continued between the 1st and 2nd waves. This is where the benefits are limited. The following will list the shortcomings that must be corrected or, if they cannot be corrected, should be noted in the limitations section of the study.

11. The author for correspondence is not indicated on the first page of the manuscript.

22. The authors did not take advantage of the study design and did not compare the studied parameters before and during the pandemic. If pre-pandemic data are published, reference should be made to these publications.

33.     Page 3, paragraph 2 below. The group of study participants is not described in sufficient detail. There are two types of shift work schedules: forward and backward shifts. It is necessary to clarify what type of shift regime was used in the study participants. If both options were used, then this indicator should be included in the model.

44.     If possible, it is necessary to describe in more detail the study participants, their nationality, education, whether they are migrants, whether there were any recoveries of COVID-19 among them, in which clinics they worked, whether they worked in the red zone (with COVID-19 patients).

55.     A significant disadvantage of the work is that the authors did not evaluate the quality of sleep using a special test, such as PSQI. This must be specified in the limitations section.

66.     Table 1 and 1st paragraph on page 6. Assessment of changes in sleep duration is not adequate. In somnology, the prevailing point of view is that there is an optimal duration of sleep, and its deviation, both downward and upward, is a deviation from the norm. Therefore, it would be more rigorous to evaluate the influence of the studied external factors on the deviation of sleep duration from the norm.

77.     Table 3 does not contain the results of the analysis corresponding to model 3 described in the methods.

Author Response

(The authors gave the same response as above.)

Reviewer 3 Report

The article presents a study conducted in healthcare field (nurses) during the pandemic picks, in Norway.

Studies on sleep quality following working shifts have been already done, but this one has been conducted in a specific sector of healthcare (nurses) during a critical period versus the previous regular ones.

What seems missing and it may be improved, at least at the discussion level and for future studies, lies in the factors considered. Taking into account this critical period of Covid 19, one should include the risk of infection of the staff in frontline and moreover the impact of these extreme factors on the staff response to the overload and these certainly may affect the quality of sleep and turnover intention or at least investigated. 

Conclusion section should be detailed in more extend and add further research on the topic. 

Author Response

(The authors gave the same response as above.)

Round 2

Reviewer 1 Report

Correction of this article are appropriate. 

Good luck.